# Ultra-ECP: Ellipse-Constrained and Point-Robust Foundation Model Adaptation for Fetal Cardiac Ultrasound Segmentation

**Minh H. N. Le**[*1,2]    D142111009@TMU.EDU.TW, JOHNMINHLE@IEEE.ORG
[1]*International Ph.D. Program in Medicine, College of Medicine, Taipei Medical University, Taipei, Taiwan;* [2]*AIBioMed Research Group, Taipei Medical University, Taipei, Taiwan*

**Khanh T. Q. Le**[*3]    22520638@GM.UIT.EDU.VN
[3]*PASSIO Lab, North Carolina A&T State University, Greensboro, NC 27411, USA*

**Tuan Vinh**[4]    TUAN.VINH@HERTFORD.OX.AC.UK
[4]*Medical Sciences Division, University of Oxford, Oxford, United Kingdom*

**Thanh-Huy Nguyen**[5]    THANHHUN@ANDREW.CMU.EDU
[5]*Computational Biology Department, Carnegie Mellon University, Pittsburgh, USA*

**Han H. Huynh**[6]    M658112001@TMU.EDU.TW
[6]*International Master Program for Translational Science, College of Medical Science and Technology, Taipei Medical University, Taipei, Taiwan*

**Khoa D. Pham**[7]    KDPHAM@AGGIES.NCAT.EDU
[7]*Industrial and Systems Engineering Department, North Carolina A&T State University, Greensboro, NC 27411, USA*

**Anh Mai Vu**[8]    MVU9@COUGARNET.UH.EDU
[8]*Department of Electrical and Computer Engineering, University of Houston, Houston, TX 77204, USA*

**Hien Q. Kha**[1,2]    D142111015@TMU.EDU.TW
[1]*International Ph.D. Program in Medicine, College of Medicine, Taipei Medical University, Taipei, Taiwan* [2]*AIBioMed Research Group, Taipei Medical University, Taipei, Taiwan*

**Phat K. Nguyen**[2,10]    M142113007@TMU.EDU.TW
[2]*AIBioMed Research Group, Taipei Medical University, Taipei, Taiwan*
[10]*International Master Program in Medicine, College of Medicine, Taipei Medical University, Taipei, Taiwan*

**Ulas Bagci**[11]    ULAS.BAGCI@NORTHWESTERN.EDU
[11]*Machine and Hybrid Intelligence Lab, Northwestern University, Chicago, IL 60611, USA.*

**Min Xu**[5,12]    MXU1@ANDREW.CMU.EDU
[5]*Computational Biology Department, Carnegie Mellon University, Pittsburgh, PA 15213, USA* [12]*Mohamed bin Zayed University of Artificial Intelligence, UAE*

**Carl Yang**[13]    J.CARLYANG@EMORY.EDU
[13]*Department of Computer Science, Emory University, Atlanta, GA 30322, USA*

**Phat K. Huynh**[7]    PKHUYNH@NCAT.EDU
[7]*Industrial and Systems Engineering Department, North Carolina A&T State University, Greensboro, NC 27411, USA*

**Nguyen Quoc Khanh Le**[2,14†]    KHANHLEE@TMU.EDU.TW
[14]*In-Service Master Program in Artificial Intelligence in Medicine, College of Medicine, Taipei Medical University, Taiwan;* [2]*AIBioMed Research Group, Taipei Medical University, Taipei, Taiwan*

LE LE VINH NGUYEN HUYNH PHAM VU KHA NGUYEN BAGCI XU YANG HUYNH LE

**Editors:** Accepted for publication at MIDL 2026

## Abstract

Accurate fetal cardiac segmentation from four-chamber ultrasound images is essential for reliable prenatal biometrics, yet foundation models such as SAM remain sensitive to point-prompt placement, produce anatomically inconsistent masks, and require costly full-model fine-tuning. We introduce **Ultra-ECP**, a parameter-efficient framework that adapts Ultra-SAM for robust single-point fetal cardiac segmentation. Ultra-ECP integrates three components: (i) a LoRA-based adaptation applied to the prompt encoder and mask decoder, reducing trainable parameters by over 98%; (ii) an Ellipse-Aware Loss that regularizes predictions toward anatomically plausible elliptical cardiac shapes; and (iii) a Point-Robust Augmentation strategy that simulates click imprecision to enhance robustness. Evaluated on the FOCUS dataset, Ultra-ECP outperforms SAM, MedSAM, and fine-tuned U-Net baselines. For thoracic segmentation, it achieves a mean DSC of **95.09%** and HD95 of **25.96 px**. For cardiac segmentation, Ultra-ECP obtains a mean DSC of **92.60%** and HD95 of **18.25 px**, while maintaining stability under point displacements of up to 10 pixels. Predictions are consistently smooth and elliptical, addressing common failure modes of existing approaches. Ultra-ECP provides an effective and computationally lightweight pathway for adapting large vision models to fetal cardiac biometrics, enabling reliable and clinically practical semi-automated tools.

**Keywords:** Fetal cardiac ultrasound, cardiac segmentation, UltraSAM, parameter-efficient learning, prenatal biometrics.

## 1. Introduction

Fetal cardiac biometric assessment relies heavily on accurate segmentation of the four-chamber view in prenatal ultrasound. Precise delineation of the cardiac chambers enables reliable measurement of cardiac dimensions, structural assessment, and the early detection of congenital abnormalities. Despite rapid advances in deep learning, automated fetal cardiac segmentation remains challenging due to image quality variability, acoustic shadowing, low contrast boundaries, and operator-dependent probe orientation (Ronneberger et al., 2015; Chen et al., 2021). Conventional CNN-based architectures such as U-Net and TransUNet often require extensive labeled data and exhibit limited generalization across acquisition settings.

Recent foundation models, particularly the Segment Anything Model (SAM) (Kirillov et al., 2023), have demonstrated strong zero-shot segmentation capabilities across diverse domains. However, their application to fetal ultrasound remains non-trivial. SAM exhibits high sensitivity to precise point-prompt localization leading to unstable performance, anatomically inconsistent predictions such as irregular boundaries or leakage into adjacent thoracic structures, and substantial computational demands when fully fine-tuned on domain-specific data (Ma et al., 2024). These limitations restrict SAM's practical use in clinical fetal cardiology workflows.

To address these challenges, we present Ultra-ECP, a parameter-efficient fine-tuning framework designed to adapt UltraSAM—a variant of SAM pretrained on medical im-

---

∗ Contributed equally
† Corresponding author

ages—for robust single-point fetal cardiac segmentation. Ultra-ECP synergistically integrates three components: Prompt- and Decoder-level LoRA adaptation enabling efficient specialization to cardiac anatomy while reducing trainable parameters by over 98%, Ellipse-Aware Loss which incorporates anatomical priors by regularizing predicted masks toward physiologically plausible elliptical shapes, and Point-Robust Augmentation simulating natural user click variability to enhance prompt robustness (Hu et al., 2022).

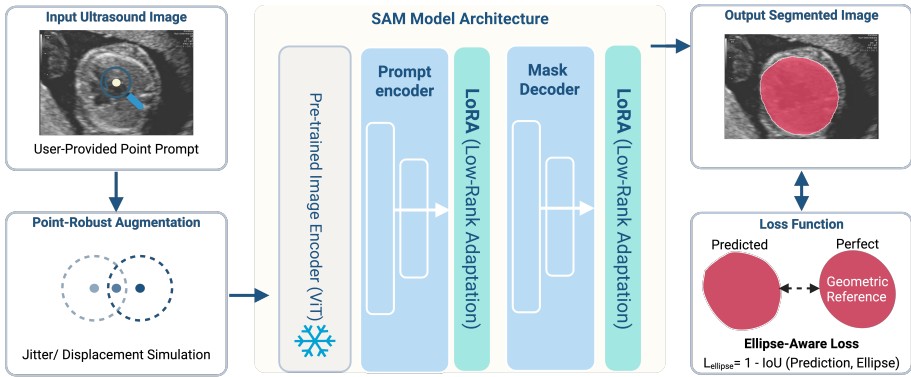

Figure 1: Overview of Ultra-ECP framework. LoRA modules are injected into UltraSAM's prompt encoder and mask decoder. Ellipse-Aware Loss regularizes predictions toward anatomical cardiac shapes, while Point-Robust Augmentation simulates clinical click imprecision.

## 2. Related Work

### 2.1. Fetal Ultrasound Segmentation

Early approaches for fetal cardiac segmentation relied on classical image processing and hand-crafted features (Carvalho et al., 2023). Deep learning methods, particularly U-Net variants, have since dominated, achieving good performance on curated datasets (Ronneberger et al., 2015). Recent works explore transformer-based architectures like TransUNet (Chen et al., 2021) and Swin-UNet (Cao et al., 2022), though these require substantial training data and struggle with domain shifts common in clinical ultrasound.

### 2.2. Segment Anything Model and Medical Adaptation

SAM (Kirillov et al., 2023) introduced promptable segmentation with a ViT image encoder, prompt encoder, and mask decoder. Medical adaptations include MedSAM (Ma et al., 2024) fine-tuned on diverse medical images and UltraSAM (Meyer et al., 2025) pretrained on ultrasound data. However, these models remain sensitive to prompt placement and produce anatomically implausible outputs in fetal ultrasound, motivating targeted adaptation.

### 2.3. Parameter-Efficient Fine-Tuning

LoRA (Hu et al., 2022) enables efficient adaptation of large models by injecting low-rank matrices into weight updates. Recent applications include vision transformers (Hu et al., 2022) and medical imaging (Shamshad et al., 2023). Anatomical priors via shape regularization (Oktay et al., 2018) and data augmentation for robustness (Shorten and Khoshgoftaar, 2019) complement PEFT approaches.

## 3. Method

### 3.1. Dataset

We evaluate Ultra-ECP on the FOCUS dataset, a publicly available collection of fetal four-chamber ultrasound images curated for cardiac biometric measurement (Wu et al., 2025). FOCUS comprises 300 second-trimester four-chamber view images acquired during routine perinatal care. Each image is manually annotated by an experienced sonographer with pixel-wise masks for the fetal thoracic cavity and cardiac chambers, enabling evaluation of both thoracic and cardiac segmentation performance. All images are de-identified and released under the CC-BY 4.0 license via Zenodo (Wu et al., 2025). We adopt the official patient-level split to avoid overlap across training, validation, and test subsets. Cardiac chamber segmentation is the primary target; thoracic segmentation serves as a secondary task.

### 3.2. Overview

Ultra-ECP adapts UltraSAM for the task of single-point fetal cardiac segmentation (Fig. 1). Given an ultrasound image $I$ and a single spatial prompt $p \in \mathbb{R}^2$, Ultra-ECP predicts a binary mask $M$ representing the cardiac chamber region. The framework introduces three key innovations that address SAM's limitations in this clinical setting: parameter-efficient fine-tuning via LoRA inserted into the prompt encoder and mask decoder, Ellipse-Aware Loss encouraging anatomically plausible outputs, and Point-Robust Augmentation improving resilience to noisy user prompts. The image encoder remains completely frozen to preserve pretrained representational power and reduce overfitting.

### 3.3. Backbone Foundation Model: UltraSAM

UltraSAM follows the standard SAM architecture comprising a ViT-based image encoder producing high-dimensional feature embeddings, a prompt encoder that embeds points, boxes, or masks, and a lightweight mask decoder fusing image and prompt embeddings into segmentation logits. Due to its strong medical-domain pretraining, UltraSAM offers a superior initialization compared to SAM (Kirillov et al., 2023). However, its prompt encoder and mask decoder are not optimized for fetal cardiac anatomy, motivating targeted adaptation.

### 3.4. Parameter-Efficient Fine-Tuning with LoRA

Full-model fine-tuning is computationally expensive and prone to overfitting, especially with limited ultrasound data. Ultra-ECP employs Low-Rank Adaptation (LoRA) (Hu et al.,

2022) to selectively adapt the most task-critical modules. Let $W$ denote a pretrained weight matrix in a transformer layer. LoRA introduces two low-rank matrices $A \in \mathbb{R}^{d \times r}$ and $B \in \mathbb{R}^{r \times k}$, such that the fine-tuned weight becomes $W' = W + BA$. Only $A$ and $B$ are optimized during training. All LoRA adapters use a low-rank configuration with rank r = 8, resulting in approximately 1.8% trainable parameters relative to the full UltraSAM model. The image encoder remains frozen. This design reduces trainable parameters by >98%, enabling training on a single consumer GPU.

### 3.5. Ellipse-Aware Loss for Anatomical Regularization

Fetal cardiac chambers in the four-chamber view exhibit a stable, elliptical morphology. Standard segmentation losses such as Dice or BCE are shape-agnostic and may produce anatomically implausible outputs. To incorporate anatomical priors, Ultra-ECP introduces Ellipse-Aware Loss $L_{\text{ellipse}}$, computed in three steps: first, fit an ellipse to the predicted binary mask $P$ using a robust least-squares ellipse fitting algorithm producing ellipse parameters $E_p$; second, generate an ellipse mask $M_e$ that fills the fitted ellipse; third, penalize deviations between $P$ and $M_e$ via $L_{\text{ellipse}} = 1 - \text{IoU}(P, M_e)$. The total training loss combines Dice segmentation loss and ellipse regularization: $L_{\text{total}} = L_{\text{seg}}(P, G) + \alpha L_{\text{ellipse}}$, where $\alpha$ controls the regularization strength (set to 1.0 in experiments). This encourages smooth, elliptical masks consistent with cardiac anatomy and mitigates noisy or leaking boundaries.

### 3.6. Point-Robust Augmentation for Clinical Usability

In real-world fetal cardiac examinations, user clicks may not precisely correspond to the cardiac center. To enhance robustness, Ultra-ECP introduces Point-Robust Augmentation simulating natural user imprecision. For each training sample, the true center $(c_x, c_y)$ of the ground-truth cardiac mask is computed, a random offset $(\Delta x, \Delta y)$ within a jitter radius $R$ is sampled, and the perturbed point $p' = (c_x + \Delta x, c_y + \Delta y)$ is used as the prompt while retaining the same ground truth mask. This forces the model to rely on contextual cardiac features rather than precise point localization, improving prompt robustness.

### 3.7. Training Details

Training uses AdamW optimizer with learning rate $1 \times 10^{-4}$. The entire image encoder remains frozen while LoRA parameters in the prompt encoder and mask decoder are trainable. Batch size is 4 with 100 epochs and early stopping. Preprocessing includes image normalization and optional contrast enhancement. Augmentation comprises horizontal flips, brightness/contrast jitter, and point jitter as described.

## 4. Experiments

### 4.1. Baselines

Ultra-ECP is compared against SAM zero-shot (Kirillov et al., 2023), MedSAM zero-shot (Ma et al., 2024), UltraSAM zero-shot, U-Net fully trained on FOCUS (Ronneberger et al., 2015), TransUNet fully trained on FOCUS (Chen et al., 2021), prompt-only fine-

tuning of SAM without LoRA or regularization, and ablation variants of Ultra-ECP including LoRA only, LoRA plus Ellipse-Aware Loss, and LoRA plus Point-Robust Augmentation.

### 4.2. Evaluation Metrics

Following prior work (Wu et al., 2025), performance is measured by Dice Similarity Coefficient (DSC), Hausdorff Distance (HD95), robustness curve under point displacements at 0, 2, 5, 10 pixels, and ellipse fitting error defined as mean squared distance between predicted mask boundary and fitted ellipse boundary.

### 4.3. Statistical Analysis

All quantitative comparisons are performed at the image level using paired statistical tests. For each method, we compute the Dice Similarity Coefficient (DSC) and 95th percentile Hausdorff Distance (HD95) per test image. To assess whether Ultra-ECP significantly outperforms baseline methods, we apply the Wilcoxon signed-rank test to paired DSC and HD95 values between Ultra-ECP and each comparator.

A two-sided significance level of $p < 0.05$ is used. When multiple pairwise comparisons are conducted, we control for the family-wise error rate using the Holm-Bonferroni procedure. In tables, statistically significant improvements of Ultra-ECP over a given baseline are marked with an asterisk. We additionally report median and interquartile range (IQR) for DSC and HD95 to better characterize performance variability across the test set.

### 4.4. Model Complexity Analysis

To quantify the efficiency of Ultra-ECP, we compare the number of trainable parameters across all methods. For UltraSAM, full fine-tuning corresponds to updating all weights in the image encoder, prompt encoder, and mask decoder. In contrast, Ultra-ECP freezes the image encoder and inserts LoRA modules only into the prompt encoder and mask decoder. As a result, Ultra-ECP optimizes less than two percent of the parameters of UltraSAM, consistent with the parameter ratios in our ablation study.

Table 1 summarizes the relative trainable parameter counts. UltraSAM full fine-tuning is normalized to one hundred percent. Ultra-ECP requires only 1.8% of these parameters, while U-Net and TransUNet fall between these extremes. This reduction in trainable parameters lowers memory consumption and simplifies optimization, making Ultra-ECP practical to train on a single commodity GPU without sacrificing segmentation accuracy.

## 5. Results

### 5.1. Quantitative Results and Qualitative Results

Quantitative results are summarized in Table 2. Ultra-ECP achieves the highest accuracy across both thoracic and cardiac tasks, outperforming SAM, MedSAM, and fully trained U-Net and TransUNet baselines, with a thoracic DSC of 95.09% and a cardiac DSC of 92.60% alongside the lowest HD95 values. These improvements highlight the effectiveness of combining LoRA-based adaptation with anatomical regularization and prompt-robust training, enabling state-of-the-art performance with less than 2% of trainable parameters.

Table 1: Relative trainable parameter counts for the evaluated models. Values are expressed as a percentage of the total parameters of fully fine-tuned UltraSAM.

| Method | Image encoder trainable | Prompt / decoder trainable | Trainable params (%) |
|---|---|---|---|
| UltraSAM (full) | Yes | Yes | 100 |
| U-Net | – | – | < 100 |
| TransUNet | – | – | < 100 |
| UltraSAM (zero-shot) | No | No | 0 |
| **Ultra-ECP (ours)** | No | Yes (LoRA only) | 1.8 |

Table 2: Quantitative comparison on FOCUS dataset. Ultra-ECP achieves state-of-the-art performance with minimal trainable parameters.

| Method | Thoracic DSC (%) | Thoracic HD95 (px) | Cardiac DSC (%) | Cardiac HD95 (px) |
|---|---|---|---|---|
| SAM (zero-shot) | 87.2 | 42.5 | 82.1 | 35.8 |
| MedSAM (zero-shot) | 89.4 | 38.2 | 84.3 | 31.2 |
| UltraSAM (zero-shot) | 91.2 | 33.1 | 87.5 | 27.4 |
| U-Net | 93.1 | 29.8 | 89.2 | 24.6 |
| TransUNet | 94.2 | 27.5 | 90.8 | 22.1 |
| **Ultra-ECP (ours)** | **95.09** | **25.96** | **92.60** | **18.25** |

Table 3 presents an ablation study quantifying the contribution of each component within Ultra-ECP. Introducing LoRA alone substantially boosts performance relative to the UltraSAM baseline, reducing HD95 from 27.4 px to 23.8 px. Adding the Ellipse-Aware Loss further improves anatomical consistency, reflected in higher DSC and reduced boundary errors. Finally, incorporating Point-Robust Augmentation yields the full Ultra-ECP model, which achieves the best overall accuracy and robustness, maintaining 91.3% DSC under 10 px prompt displacement while requiring only 1.8% trainable parameters.

Table 3: Ablation study showing contributions of each Ultra-ECP component.

| Variant | Cardiac DSC (%) | HD95 (px) | Param. (%) | Robustness (10px) |
|---|---|---|---|---|
| UltraSAM baseline | 87.5 | 27.4 | 100 | 78.2 |
| + LoRA | 90.1 | 23.8 | 1.8 | 82.4 |
| + Ellipse Loss | 91.7 | 20.5 | 1.8 | 85.1 |
| + Point Aug | **92.60** | **18.25** | 1.8 | **91.3** |

Performance remains stable under prompt displacement up to 10 px (Fig. 2). Compared to SAM/MedSAM, Ultra-ECP yields superior anatomical consistency and robustness. Compared to U-Net, Ultra-ECP delivers higher accuracy with <2% trainable parameters.

Predicted masks are smooth, continuous, and closely aligned with cardiac boundaries (Fig. 3). Ellipse regularization reduces boundary noise, and point jitter training prevents the collapse or drift commonly observed in SAM-based methods.

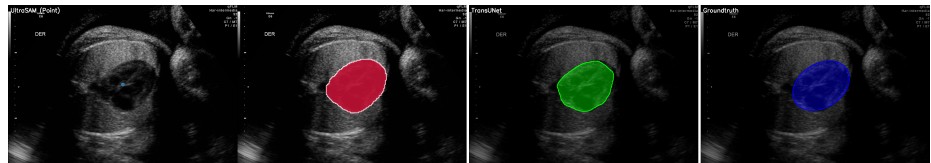

Figure 2: Robustness to point displacement. Ultra-ECP maintains stable DSC across 0-10 pixel perturbations.

Figure 3: Qualitative comparison. Ultra-ECP produces smooth, anatomically consistent cardiac masks even with noisy prompts, unlike SAM baselines.

## 6. Discussion

Ultra-ECP demonstrates that large vision foundation models can be effectively adapted to specialized fetal ultrasound tasks through parameter-efficient fine-tuning with anatomical priors. By injecting LoRA modules into UltraSAM's prompt encoder and mask decoder while keeping the image encoder frozen, the framework learns cardiac-specific representations with less than two percent of trainable parameters, yet achieves state-of-the-art performance on the FOCUS dataset for both thoracic and cardiac segmentation.

The Ellipse-Aware Loss enforces anatomical plausibility by penalizing deviations between predicted masks and fitted ellipses. Unlike shape-agnostic losses such as Dice or cross-entropy, this regularization encourages smooth, elliptical cardiac shapes consistent with four-chamber view morphology, reducing boundary noise and mitigating leakage into adjacent thoracic structures. Point-Robust Augmentation reduces sensitivity to prompt placement, a key concern in clinical practice where sonographers may only roughly indicate the cardiac chamber. Ultra-ECP maintains stable performance under perturbations of up to ten pixels, whereas zero-shot SAM and MedSAM degrade markedly—an important property for practical deployment under time constraints and probe motion.

Compared with U-Net and TransUNet, Ultra-ECP offers a favorable accuracy-efficiency trade-off, suggesting that parameter-efficient fine-tuning with domain-specific priors is a promising alternative to training task-specific models from scratch.

### 6.1. Limitations and Future Work

Ellipse-Aware Loss relies on robust ellipse fitting and may introduce bias when acoustic shadowing, signal dropout, or highly irregular contours prevent accurate fitting (Fig. 4). This regularization is intended as a soft anatomical prior for normal biometrics and should not be interpreted as enforcing normal anatomy in congenital heart disease, where car-

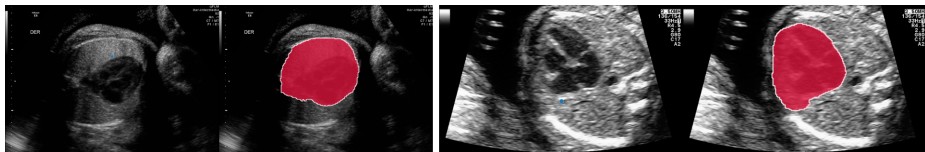

Figure 4: Some qualitative examples illustrating failure cases of ellipse fitting

diac morphology may deviate substantially from elliptical assumptions. Additionally, we consider only single-point prompts; multi-point or box prompts could improve controllability. Our evaluation is restricted to a single four-chamber dataset, and domain shifts across scanners, institutions, and gestational ages may affect generalization. Future work will explore multi-center cohorts, temporal consistency across cine sequences, and extension to three-dimensional fetal echocardiography.

### 6.2. Clinical Implications

Accurate fetal cardiac segmentation underpins reliable cardiothoracic ratio measurement for congenital heart disease screening. By stabilizing segmentation under imperfect prompts with minimal annotation burden, Ultra-ECP can support semi-automated workflows in routine obstetric ultrasound, enabling near real-time cardiac mask visualization while keeping the clinician in the loop.

## 7. Conclusion

We presented Ultra-ECP, a parameter-efficient, anatomically informed framework for adapting UltraSAM to fetal cardiac segmentation. Ultra-ECP achieves robust, high-accuracy segmentation from a single point prompt while requiring minimal trainable parameters. The framework significantly improves anatomical consistency and prompt robustness, enabling reliable and clinically deployable fetal cardiac biometric tools. Future work will explore multi-view consistency, temporal modeling across cine sequences, and extension to broader prenatal structural assessments.

### Ethics Statement

This study uses only the publicly available, de-identified FOCUS dataset (Wu et al., 2025). No new patient data were collected and no IRB approval was required. The proposed method is intended as a decision-support tool and is not designed to replace expert clinical judgment.

### Acknowledgments

We thank the FOCUS dataset creators for public data release. This work was supported by the National Science and Technology Council, Taiwan [grant number NSTC114- 2221-E-038-015].

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
