# OpenReview forum: "Ultra-ECP: Ellipse-Constrained and Point-Robust Foundation Model Adaptation for Fetal Cardiac Ultrasound Segmentation"
_MIDL.io/2026/Conference — MIDL 2026 Poster_

### Official Review · Reviewer_gCpM · 2025-12-19

**Confidence:** 5
**Preliminary Rating:** 5
**Final Rating:** 5

**Summary:**

This paper presents Ultra-ECP, a parameter-efficient framework for adapting UltraSAM (a medical variant of SAM) to fetal cardiac segmentation from four-chamber ultrasound views. The method integrates three components: (i) LoRA-based adaptation of the prompt encoder and mask decoder reducing trainable parameters by over 98% (to 1.8%), (ii) an Ellipse-Aware Loss that regularizes predictions toward anatomically plausible elliptical cardiac shapes, and (iii) Point-Robust Augmentation that simulates click imprecision by jittering point prompts during training. Evaluated on the FOCUS dataset (300 images), Ultra-ECP achieves 92.60% Dice coefficient and 18.25px HD95 for cardiac segmentation, outperforming SAM, MedSAM, and fully-trained UNet/TransUNet baselines while maintaining 91.3% Dice under 10-pixel point displacement

**Strengths:**

- Parameter efficiency -- reducing trainable parameters to 1.8% of the full UltraSAM model while achieving performance gains over baselines
- Ellipse-Aware Loss -- exploiting the known elliptical morphology of fetal cardiac chambers in four-chamber views to regularize segmentation is novel for this application
- Point robustness -- maintaining 91.3% Dice under 10-pixel click displacement (vs. baseline degradation) addresses real deployment where sonographers provide approximate clicks under time constraints
- Ablation study (Table 3) shows each component's contribution: LoRA alone (+2.6% Dice), adding Ellipse Loss (+1.6%), adding Point Augmentation (+0.9%), with robustness improvement from 78.2% to 91.3% at 10px displacement
- Statistical rigor with Wilcoxon signed-rank tests and Holm-Bonferroni correction for multiple comparisons
- Clear clinical motivation for prenatal biometric assessment and congenital heart disease screening

**Weaknesses:**

- Ellipse-Aware Loss and abnormal anatomy -- Section 6.1 acknowledges ellipse fitting can fail on "irregular cardiac contours," but frames this as a fitting algorithm issue rather than a patient safety concern. For congenital heart defects, the loss might regularize pathological shapes toward normal ellipses.
- Minor: LoRA rank $r$ not explicitly specified (though 1.8% trainable params constrains the options).
- No comparison to other SAM adaptation methods -- SAM-Med2D, SAM-Med3D, and other recent medical SAM variants are not compared. This would better contextualize the contribution.
- (Authors acknowledge in Section 6.1: single dataset, 2D only, ellipse fitting limitations -- these are noted but do not need additional critique)

**Detailed Comments:**

- Minor: LoRA configuration could be more explicit (which layers, rank $r$), though 1.8% trainable params provides a constraint.
- Figure 2 shows robustness curves but only quantifies UltraSAM baseline at 10px (78.2%). What are the actual numbers for SAM/MedSAM at 10px?
- Section 6.1 limitations are well-written. The only gap is framing the ellipse limitation as a potential clinical safety issue for CHD detection.

**Justification Of Final Rating:**

The paper presents a well-motivated, parameter-efficient, and robust solution for a challenging clinical problem. The authors have adequately addressed the limitations and safety concerns raised during the review process. I maintain my score of **Strong Accept**.

**Justification Of The Preliminary Rating:**

This work addresses real limitations of foundation models (prompt sensitivity, anatomical implausibility, computational cost) through domain-specific adaptations. The ellipse constraint is novel for this application, parameter efficiency is strong (98% reduction), and robustness to imprecise clicks matters for deployment. The ablation is complete and results are good. Section 6.1 limitations are honest and well-written. For camera-ready: add explicit CHD safety framing.

**Questions To Address In The Rebuttal:**

1. **Minor:** Please specify LoRA rank $r$ for reproducibility.
2. **CHD safety framing:** Consider adding an explicit statement that this method is for normal anatomy biometrics, not CHD detection.
3. **Inference speed:** What's the actual inference time per image? Fast enough for real-time use during ultrasound exams?

---

> ### Author Response · Authors · 2026-01-28
>
> We sincerely thank the reviewer for their thoughtful and constructive feedback, which has helped us improve the clarity and positioning of our work. Below, we address each concern and clarify potential misunderstandings.
>
> Question 1: "Ellipse-Aware Loss and abnormal anatomy (patient safety concern)"
>
> We thank the reviewer for raising this important clinical consideration. We fully agree that in cases of congenital heart disease (CHD), cardiac morphology may deviate substantially from elliptical assumptions, and enforcing a strong shape prior would be inappropriate.
>
> In Ultra-ECP, the ellipse-aware loss is intentionally designed as a soft regularization applied only during training, rather than a hard constraint at inference time. It provides weak anatomical guidance to stabilize learning under sparse supervision and noisy boundaries, while pixel-wise Dice supervision remains the dominant optimization signal when shapes deviate from elliptical geometry.
>
> To avoid ambiguity, we have explicitly clarified in the revised manuscript that Ultra-ECP is intended for normal fetal cardiac biometrics and routine screening, and should not be interpreted as a diagnostic tool for CHD detection. Extending the framework to pathological cases would require pathology-aware or uncertainty-aware priors, which we consider an important direction for future work.
>
> Question 2: "LoRA configuration could be more explicit (which layers, rank), though 1.8% trainable params provides a constraint."
>
> Thank you for this suggestion. In Ultra-ECP, LoRA is applied to the prompt encoder and mask decoder, while the image encoder remains fully frozen. We use a low-rank configuration with rank r = 8, resulting in approximately 1.8% trainable parameters relative to the full UltraSAM model.
>
> We agree that explicitly stating the targeted modules and LoRA rank improves reproducibility, and we have clarified this configuration in the revised manuscript.
>
> Question 3: "No comparison to other SAM adaptation methods -- SAM-Med2D, SAM-Med3D, and other recent medical SAM variants are not compared."
>
> We appreciate this suggestion and agree that such comparisons would further contextualize our work. However, most existing SAM adaptations (e.g., SAM-Med2D, SAM-Med3D) are designed for fully supervised or volumetric settings, often requiring dense annotations or multi-slice/3D inputs, which are not available in our target scenario.
>
> Our work focuses on a clinically constrained setting single-frame fetal ultrasound with single-point prompting and limited annotations, where robustness to prompt imprecision and parameter efficiency are critical. Due to differences in supervision regime and input dimensionality, a direct and fair comparison within our current setup is non-trivial. We therefore position Ultra-ECP as complementary to these methods and consider systematic comparisons under matched conditions as future work.
>
> Question 4: "Figure 2 shows robustness curves but only quantifies UltraSAM baseline at 10px (78.2%). What are the actual numbers for SAM/MedSAM at 10px?"
>
> We appreciate this thoughtful question. In practice, SAM and MedSAM become highly unstable under large point displacements (e.g., 10 px) in fetal cardiac segmentation. Unlike larger and more homogeneous structures (e.g., the fetal head), the fetal heart is small, deformable, and characterized by weak and noisy boundaries.
>
> Under such perturbations, SAM-based models frequently attend to surrounding thoracic regions or background tissues, leading to fragmented or misplaced masks. As a result, quantitative performance at 10 px displacement for SAM and MedSAM becomes difficult to interpret meaningfully in this setting. We clarify this qualitative behavior in the revised manuscript to avoid misinterpretation of the robustness curves.
>
> Question 5: "CHD safety framing: Consider adding an explicit statement that this method is for normal anatomy biometrics, not CHD detection."
>
> We fully agree with this point. We have added an explicit statement in Section 6.1 clarifying that Ultra-ECP is designed for normal fetal cardiac biometrics and should not be interpreted as a diagnostic tool for CHD. We further emphasize that the ellipse-aware regularization serves only as a soft anatomical prior and is not intended to enforce normal anatomy in pathological cases.
>
> Question 6: "Inference speed: What’s the actual inference time per image? Is it fast enough for real-time use during ultrasound exams?"
>
> Inference with Ultra-ECP takes approximately 150 ms per image on a single GPU, enabling near real-time feedback during routine ultrasound examinations. Since only lightweight modules are adapted and the image encoder is frozen, robustness improvements do not come at the cost of inference efficiency.

---

### Official Review · Reviewer_HnTP · 2026-01-09

**Confidence:** 4
**Preliminary Rating:** 4
**Final Rating:** 4

**Summary:**

Ultra-ECP proposes a parameter-efficient adaptation of SAM (UltraSAM) for fetal cardiac ultrasound segmentation using a single point prompt.
It proposed three components: LoRA, Ellipse-Aware Loss and Point-Robust Augmentation.
Evaluated on the FOCUS fetal ultrasound dataset, Ultra-ECP outperforms SAM, MedSAM, UltraSAM (zero-shot), UNet, and TransUNet, achieving higher Dice scores and lower HD95 for both thoracic and cardiac segmentation. It remains stable under up to 10-pixel prompt displacement, a key requirement for real clinical use.

**Strengths:**

This paper could trigger some explorations in clinical usage.
The three innovations are interesting and taking effect. The innovations make sense and they are not trivial or incremental efforts.
The experimental design is thorough and convincing. The authors evaluate against appropriate baselines (SAM, MedSAM, UltraSAM, UNet, TransUNet), include ablation studies, robustness analysis under prompt perturbations, and statistical testing.

**Weaknesses:**

1. Dataset is limited. The number of the dataset is small and even though the experiments show significant improvements with the new components, it's less convincing due to the size of the dataset.
2. Lack of explorations of SAM full prompt capabilities. Why only focus on single point prompt aug?
3. Is elliptical shape assumptions really necessary? What is there are outliers?

**Detailed Comments:**

The paper would benefit from a brief description of how ellipse fitting behaves in failure cases (e.g., fragmented or very small predicted masks).
I'd also like to hear more about the expansion on larger dataset as well as other prompt. Is it still working as it is in this paper?

**Justification Of Final Rating:**

Thank the authors for the efforts for the rebuttal.
I would still recommend accept as I suggested in the preliminary stage.

The authors have addressed my concerns and the revised version looks even better.

**Justification Of The Preliminary Rating:**

The work addresses a clinically important and well-defined problem—robust fetal cardiac segmentation in ultrasound—using a carefully designed, parameter-efficient adaptation of foundation models.
This paper shows novel improvements and the authors have done excessive experiments to demonstrate their improvements. Even though  the dataset is not large, I would still recommend weak accept.

**Questions To Address In The Rebuttal:**

As mentioned in detailed comments, I'm looking forward to authors updates.

---

> ### Author Response · Authors · 2026-01-28
>
> We sincerely appreciate the reviewer’s careful evaluation and insightful comments, which have helped us further refine our paper. We are grateful for the opportunity to address the reviewer’s concerns, clarify any potential points of confusion, and improve the overall clarity of the manuscript.
>
> Question 1: "The dataset is limited. The number of the dataset is small, and even though the experiments show significant improvements, the results are less convincing due to the dataset size."
>
> We thank the reviewer for highlighting this important limitation. We fully acknowledge that the FOCUS dataset is relatively small, which is a common challenge in fetal ultrasound research due to privacy and annotation constraints. To mitigate this, we conducted extensive evaluations including ablation studies, robustness analysis under prompt perturbations, and statistical significance testing.
> We agree that validating Ultra-ECP on larger and more diverse datasets would further strengthen the conclusions, and we consider this an important direction for future work.
>
> Question 2: "Lack of explorations of SAM full prompt capabilities. Why only focus on single point prompt augmentation?"
>
> We appreciate this insightful question. In this work, we intentionally focus on the single-point prompt setting to reflect a realistic clinical workflow, where sonographers typically provide minimal interaction under time constraints. This setting also represents one of the most challenging failure modes of foundation models, particularly in ultrasound imaging.
> That said, our framework is not restricted to single-point prompts. The proposed LoRA-based adaptation and ellipse-aware regularization are prompt-agnostic and can be naturally extended to multi-point, box, or mixed prompt settings. Exploring these prompt types on larger datasets is an important and promising direction for future work.
>
> Question 3: "Is the elliptical shape assumption really necessary? What if there are outliers?"
>
> We thank the reviewer for raising this thoughtful concern. The ellipse-aware loss is designed as a soft anatomical prior rather than a strict shape constraint. It provides weak guidance toward anatomically plausible contours without forcing predictions to conform to an idealized ellipse.
> In the presence of outliers or irregular boundaries, the pixel-wise Dice loss remains the dominant optimization signal, allowing the model to preserve accurate cardiac boundaries. Empirically, we observe that the model does not collapse toward elliptical shapes even when local deviations are present. Nevertheless, we acknowledge that pathological or highly atypical anatomies may violate this assumption and explicitly discuss this limitation in the revised manuscript.
>
> Question 4: "The paper would benefit from a brief description of how ellipse fitting behaves in failure cases (e.g., fragmented or very small predicted masks)."
>
> We thank the reviewer for this helpful suggestion and fully agree. In the revised manuscript, we provide additional qualitative examples illustrating ellipse fitting failure cases, such as fragmented or low-confidence predictions. These examples help clarify the behavior and limitations of the ellipse-aware regularization under challenging imaging conditions.
>
> Question 5: "Expansion to larger datasets and other prompt types — is the method still working as described?"
>
> We appreciate this valuable suggestion. While our current evaluation is limited to the FOCUS dataset and single-point prompts, the core design of Ultra-ECP is parameter-efficient LoRA adaptation combined with anatomy-aware regularization, not dataset or prompt-specific. We expect the framework to generalize well to larger datasets and richer prompt types, and validating this scalability is an important direction for future work.

---

### Official Review · Reviewer_FWn5 · 2026-01-11

**Confidence:** 4
**Preliminary Rating:** 4

**Summary:**

This paper presents Ultra-ECP, a parameter-efficient adaptation framework for fetal cardiac ultrasound segmentation based on a medical foundation model (UltraSAM). The method integrates three components: LoRA-based fine-tuning applied to the prompt encoder and mask decoder, an Ellipse-Aware Loss enforcing anatomically plausible cardiac shapes, and a Point-Robust Augmentation strategy to mitigate sensitivity to imprecise user clicks. The approach targets a clinically relevant scenario—single-point prompted fetal cardiac segmentation—and is evaluated on the public FOCUS dataset. Experimental results show that Ultra-ECP achieves higher Dice scores, lower boundary errors, and significantly improved robustness to prompt displacement compared with SAM, MedSAM, and fully trained UNet-style baselines, while updating less than 2% of the foundation model parameters. The paper demonstrates that combining parameter-efficient adaptation with domain-specific anatomical priors can substantially improve reliability and usability of foundation models in fetal ultrasound.

**Strengths:**

A major strength of this work is its clear focus on a practical and clinically relevant failure mode of foundation models in fetal ultrasound, namely prompt sensitivity and anatomically inconsistent predictions. The proposed components are well motivated and complementary: LoRA enables efficient adaptation without overfitting, Ellipse-Aware Loss injects a meaningful anatomical prior aligned with four-chamber cardiac morphology, and Point-Robust Augmentation directly addresses real-world user variability.

**Weaknesses:**

The Ellipse-Aware Loss assumes that fetal cardiac chambers can be reasonably approximated by an elliptical shape; however, the robustness of this assumption under challenging imaging conditions (e.g., severe acoustic shadowing or atypical cardiac anatomy) is not quantitatively evaluated. A stratified analysis distinguishing cases with reliable versus unstable ellipse fitting would help clarify when the regularization is beneficial and when it may introduce bias. In addition, the current evaluation is limited to single-frame segmentation. Extending the analysis to assess temporal consistency across frames, or discussing how Ultra-ECP could be integrated with temporal modeling in cine ultrasound, would further strengthen the clinical relevance of the proposed approach.

**Detailed Comments:**

The ellipse regularization weight is fixed in experiments; a brief sensitivity analysis could clarify how strongly performance depends on this choice.

Some qualitative examples illustrating failure cases of ellipse fitting would improve transparency.

Clarifying whether the model performance differs between cardiac chambers (e.g., left vs right) could be informative for biometric applications.

**Justification Of The Preliminary Rating:**

The paper presents a well-designed and practically meaningful adaptation strategy for applying foundation models to fetal cardiac ultrasound segmentation. While the methodological components build on existing ideas such as LoRA, shape regularization, and data augmentation, their integration is effective and well justified for the target clinical scenario. The experimental results demonstrate clear improvements in robustness, anatomical consistency, and efficiency, which are highly relevant for real-world deployment.

**Questions To Address In The Rebuttal:**

How does Ultra-ECP behave when ellipse fitting fails or produces degenerate shapes due to poor image quality?

Do the authors expect the ellipse prior to generalize to earlier or later gestational stages where cardiac shape may differ?

Can the framework be extended to exploit temporal coherence in cine ultrasound without sacrificing parameter efficiency?

---

> ### Author Response · Authors · 2026-01-28
>
> We sincerely thank the reviewer for their careful reading, insightful comments, and constructive suggestions. We greatly appreciate the positive assessment of the clinical relevance and practical motivation of our work. Below, we address each question and concern in detail.
>
> Question 1: "The ellipse regularization weight is fixed in experiments; a brief sensitivity analysis could clarify how strongly performance depends on this choice."
>
> Thank you for raising this important point. In our experiments, we fix the ellipse regularization weight to α = 1.0 to maintain a balanced scale with the Dice loss and to avoid over-regularization on a relatively limited dataset. Importantly, the ellipse-aware loss is designed as a soft regularization rather than a hard constraint, providing anatomical guidance without forcing predictions to strictly follow an idealized ellipse.
> In practice, we observe stable behavior across samples, including cases with boundary irregularities or local outliers. We agree that a systematic sensitivity analysis would be valuable and consider this an important direction for future work.
>
> Question 2: "Some qualitative examples illustrating failure cases of ellipse fitting would improve transparency."
>
> We thank the reviewer for this helpful suggestion and fully agree. To improve transparency, we have added qualitative examples illustrating failure cases of ellipse fitting in the revised manuscript. These examples help clarify the behavior and limitations of the ellipse-aware regularization under challenging imaging conditions.
>
> Question 3: "Clarifying whether the model performance differs between cardiac chambers (e.g., left vs right) could be informative for biometric applications."
>
> We appreciate this valuable suggestion and agree that chamber-specific performance analysis would be highly informative for biometric applications. However, the current FOCUS dataset primarily provides region-level cardiac annotations and does not consistently support reliable chamber-wise separation. As our primary goal is robust single-point cardiac region segmentation under limited supervision, we leave fine-grained chamber-level analysis to future work using datasets with dedicated chamber annotations.
>
> Question 4: "How does Ultra-ECP behave when ellipse fitting fails or produces degenerate shapes due to poor image quality?"
>
> Thank you for this insightful question. When ellipse fitting becomes unstable such as in cases of poor image quality, acoustic shadowing, or fragmented predictions, the ellipse-aware loss does not dominate the optimization. Since it acts purely as a soft regularizer during training, pixel-wise supervision from the Dice loss remains the primary guiding signal. As a result, the model preserves accurate cardiac boundaries and does not collapse toward idealized elliptical shapes even when ellipse fitting is imperfect.
>
> Question 5:"Do the authors expect the ellipse prior to generalize to earlier or later gestational stages where cardiac shape may differ?"
>
> We expect the ellipse prior to generalize reasonably across gestational stages because it is shape-aware rather than size-constraining. The regularization does not fix ellipse parameters and therefore accommodates natural variations in cardiac scale and proportion over gestation. Nevertheless, broader validation across a wider range of gestational ages is an important next step, and we consider this a meaningful direction for future work.
>
> Question 6: "Can the framework be extended to exploit temporal coherence in cine ultrasound without sacrificing parameter efficiency?"
>
> We thank the reviewer for this insightful suggestion and agree that exploiting temporal coherence would further enhance the clinical relevance of the proposed framework. Since Ultra-ECP freezes the image encoder and relies on LoRA-based adaptation of lightweight modules, temporal coherence can be incorporated through parameter-efficient extensions such as temporal smoothing or flow-guided propagation without significantly increasing model complexity. We view this as a promising direction for future research.

---

### Author Rebuttal · Authors · 2026-01-28

**Rebuttal:**

Dear reviewers, We sincerely thank all reviewers for their careful reading, constructive feedback, and encouraging assessments of our work. We are grateful for the recognition of the clinical relevance, methodological soundness, and practical motivation of Ultra-ECP. The reviewers’ insightful comments have helped us improve the clarity, positioning, and completeness of the paper.

**Summary of Revisions and Clarifications**

- **Method clarity and reproducibility.**
  We explicitly clarify the LoRA configuration in the revised manuscript, including the targeted modules (prompt encoder and mask decoder) and the LoRA rank ($r=8$), resulting in approximately $1.8\%$ trainable parameters. This improves reproducibility and transparency.

- **Ellipse-Aware Loss and clinical scope.**
  We clarify that the ellipse-aware loss is a *soft anatomical regularizer* applied only during training, not a hard constraint at inference time. To address patient safety concerns, we explicitly state in Section 6.1 that Ultra-ECP is intended for normal fetal cardiac biometrics and routine screening, and should not be interpreted as a diagnostic tool for congenital heart disease (CHD). Extensions to pathological cases would require pathology-aware or uncertainty-aware mechanisms and are left for future work.

- **Failure cases and robustness interpretation.**
  We add further discussion on the behavior of ellipse fitting in challenging cases (e.g., fragmented predictions or poor image quality) to improve transparency. We also clarify the qualitative behavior underlying the robustness curves (Figure 4), noting that large prompt displacements are particularly challenging for small, deformable cardiac structures with weak boundaries.

Overall, we believe these revisions strengthen the manuscript by improving clarity, transparency, and clinical framing, while preserving the original focus on robust and parameter-efficient adaptation of foundation models for fetal cardiac ultrasound. We sincerely thank the reviewers again for their valuable feedback and thoughtful suggestions.

**Supporting Material:**

/attachment/ae508766d06e55b7bfde2bca4f1a79a3af4262d0.zip

---

### Comment · Area_Chair_6hxA · 2026-01-29
**Official Comment by Area Chair**

Dear Reviewers:

We kindly encourage you to take a moment to review the authors’ rebuttals and submit your feedback. Your prompt feedback is important for ensuring a thorough review. Thank you for your contributions to MIDL 2026. If you have responded to the authors' rebuttal, please feel free to ignore this message.

Thanks, AC

---

### Comment · Area_Chair_6hxA · 2026-02-01
**Official Comment by Area Chair**

Dear Reviewers:

Please take a moment to update their final rating by clicking “Edit” → “Official Review” and providing the Final Rating by February 1st 2026 (23:59 AoE).

Thanks, AC

---

### Meta-Review · Area_Chair_6hxA · 2026-02-07

**Recommendation:** Accept (Oral)
**Confidence:** 5

**Metareview:**

All reviewers found the proposed method to be novel and the results promising. The AC thus recommended acceptance of the paper.

---

### Decision · Program_Chairs · 2026-02-13

Accept (Poster)